# Doping with Rare Earth Elements and Loading Cocatalysts to Improve the Solar Water Splitting Performance of BiVO$_4$

**Meng Wang [1], Lan Wu [1,*], Feng Zhang [2], Lili Gao [3], Lei Geng [1], Jiabao Ge [1], Kaige Tian [1], Huan Chai [3], Huilin Niu [3], Yang Liu [1] and Jun Jin [3,*]**

[1] College of Chemical Engineering, Northwest Minzu University, Lanzhou 730030, China; mengwang_0516@163.com (M.W.); gengl1214@163.com (L.G.); 17789356894@163.com (J.G.); 13526287931@163.com (K.T.)

[2] Lanzhou Petrochemical Research Center, Petrochemical Research Institute, Petrochina, Lanzhou 730060, China; zhangfeng81@petrochina.com.cn

[3] State Key Laboratory of Applied Organic Chemistry (SKLAOC), The Key Laboratory of Catalytic Engineering of Gansu Province, College of Chemistry and Chemical Engineering, Lanzhou University, Lanzhou 730000, China; gaoll21@lzu.edu.cn (L.G.); chaih20@lzu.edu.cn (H.C.)

* Correspondence: wulan@xbmu.edu.cn (L.W.); jinjun@lzu.edu.cn (J.J.)

**Abstract:** BiVO$_4$ is a highly promising material for Photoelectrochemical (PEC) water splitting photoanodes due to its narrow band gap value (~2.4 eV) and its ability to efficiently absorb visible light. However, the short hole migration distance, severe surface complexation, and low carrier separation efficiency limit its application. Therefore, in this paper, BiVO$_4$ was modified by loading CoOOH cocatalyst on the rare earth element Nd-doped BiVO$_4$ (Nd-BiVO$_4$) photoanode. The physical characterization and electrochemical test results showed that Nd doping will cause lattice distortion of BiVO$_4$ and introduce impurity energy levels to capture electrons to increase carrier concentration, thereby improving carrier separation efficiency. Further loading of surface CoOOH cocatalyst can accelerate charge separation and inhibit electron–hole recombination. Ultimately, the prepared target photoanode (CoOOH-Nd-BiVO$_4$) exhibits an excellent photocurrent density (2.4 mAcm$^{-2}$) at 1.23 V versus reversible hydrogen electrode potential (vs. RHE), which is 2.67 times higher than that of pure BiVO$_4$ (0.9 mA cm$^{-2}$), and the onset potential is negatively shifted by 214 mV. The formation of the internal energy states of rare earth metal elements can reduce the photoexcited electron–hole pair recombination, so as to achieve efficient photochemical water decomposition ability. CoOOH is an efficient and suitable oxygen evolution cocatalyst (OEC), and OEC decoration of BiVO$_4$ surface is of great significance for inhibiting surface charge recombination. This work provides a new strategy for achieving effective PEC water oxidation of BiVO$_4$.

**Keywords:** BiVO$_4$; doping; rare earth elements; cocatalysts; photoelectrochemical water splitting





## 1. Introduction

In the contemporary world, the exhaustion of fossil energy and environmental pollution are major problems, and it is urgent to find renewable energy to replace fossil fuels [1–4]. Photoelectrochemical (PEC) hydrogen production is a promising method for hydrogen production, which is green and has low energy consumption, and has received wide attention because it can effectively utilize solar energy and electricity to achieve hydrogen production by water splitting [5–8]. However, the efficiency of PEC water splitting is limited owing to the slow kinetics of the four-electron step water oxidation reaction in the hydrolysis process [9–11]. Therefore, there is a need to develop a cheap and highly active photoanode material.

Among the types of semiconductors used to build photoanodes, metal oxide semiconductors (TiO$_2$ [12,13], WO$_3$ [14,15], Fe$_2$O$_3$ [16,17], ZnO [18,19], BiVO$_4$ [20,21], etc.) have received a lot of attention from researchers. Both TiO$_2$ and ZnO have wide energy band

gaps (>3.0 eV), which make them only absorb ultraviolet light, resulting in extremely low solar energy utilization and hydrogen production efficiency [22,23]. $Fe_2O_3$ has poor conductivity, a low light absorption coefficient, a short hole diffusion length, and poor surface oxygen evolution kinetics [24]. $WO_3$ has poor light absorption ability and is thermodynamically unstable in the electrolyte, which is susceptible to photocorrosion caused by peroxide species generated during water oxidation [25]. $BiVO_4$ is also a metal oxide semiconductor material. Because it is composed of relatively abundant elements in the earth, it is an *n*-type semiconductor material with a band gap value of about 2.4 eV [26,27]. It has a band edge position suitable for oxygen evolution reaction (OER), which can effectively absorb visible light and become a potential photoanode material [28,29]. However, due to the short hole diffusion distance, low electron mobility, and poor water oxidation reaction kinetics of $BiVO_4$ photoanode, photogenerated electrons and holes are easily recombined, resulting in low hole utilization [30–32]. The water splitting performance of PEC is limited because most of the holes recombine rapidly when migrating to the surface, resulting in poor PEC performance of $BiVO_4$.

Therefore, doping of heteroatoms [20], construction of heterojunctions [33], introduction of surface cocatalysts [34], and morphology modulation [35] are used to solve the above-mentioned problems. Usually, metal doping is one of the most effective ways because metal ion doping can introduce impurity energy levels in the semiconductor and promote the separation of photogenerated carriers [36]. With the continuous development of science and technology, the use of earth-rich, cheap, and effective rare earth element photocatalysts has been continuously developed. This will create the opportunity to replace rare and precious metals [37]. Rare earth elements have a special 4f orbital, and their ion configuration is $4f^n5s^25p^6$. Therefore, this makes them have unique chemical properties and a wider space for use [38]. In particular, rare earth elements are highly favored in photocatalysis due to their abundant energy levels and special 4f electron leap properties [39]. In recent years, literature has shown that rare earth ions (Gd [27], Sm/Tm [40], Eu [41], etc.) can be used as active cocatalysts and dopants. Also, other literature has reported the enhanced photocatalytic performance of bismuth vanadate doped with rare earth elements. Among them, Liu et al. and Luo et al. reported the lanthanide-doped tetragonal zircon phase bismuth vanadate. Moreover, the Nd, Sm, Gd, and Yb-doped bismuth vanadate were prepared by hydrothermal method, and the doping of lanthanide elements made the samples appear to possess tetragonal zircon phase, which effectively enhanced the photocatalytic activity of bismuth vanadate samples [42,43]. Umesh Prasad et al. reported a $BiVO_4$ doped with Er, W, and constructed a heterojunction photoanode with $WO_3$, namely $WO_3$/(Er, W): $BiVO_4$ photoanode, which enhanced PEC performance. The rare earth element Er-doped $BiVO_4$ can improve the bulk charge separation efficiency, thereby improving the conductivity of the charge carriers [44].

Similarly, the modification of surface cocatalysts can effectively inhibit the charge recombination on the $BiVO_4$ surface, which can increase the surface reaction kinetics and reduce the overpotential of the reaction, thus improving the hole transfer efficiency [45–48]. For example, Wang et al. successfully loaded the F-doped FeOOH on $BiVO_4$ by a one-step hydrothermal method, which promoted the transfer of photogenerated carriers and the separation of electrons and holes [49]. In addition, CoOOH and NiOOH cocatalysts, which can also improve the hole transfer efficiency and enhance the photocurrent, were also introduced into the design of $BiVO_4$ photoanodes [50,51].

In this paper, a new composite photoanode was constructed using a simple and time-saving method to prepare a novel CoOOH-Nd-$BiVO_4$ photoanode by doping rare earth elements (Nd) in $BiVO_4$ to increase the carrier density. Nd has a special 4f electron leap property, which can trap electrons and improve the hole separation efficiency. Then the CoOOH cocatalyst was loaded on the surface of Nd-$BiVO_4$ by hydrothermal method. Cocatalyst can accelerate the charge separation and suppress the electron–hole recombination. The prepared CoOOH-Nd-$BiVO_4$ composite photoanode has excellent PEC performance. The photocurrent density of the best nanocomposite photoanode is 2.4 mA $cm^{-2}$ at

1.23 V vs. RHE, which is 2.67 times compared with the pure $BiVO_4$ (0.9 mA cm$^{-2}$), and the onset potential is a 214 mV negative shift. The composite photoanode has fast hole transfer kinetic properties, which effectively inhibits the recombination of photogenerated carriers and improves the PEC performance. This paper provides a new strategy to improve the PEC performance of the photoanode and accelerate the kinetics of water oxidation.

## 2. Results and Discussion

The microscopic surface morphology of all photoanode materials was investigated using field emission scanning electron microscopy (SEM). Figure 1a shows the pristine BiOI films in the shape of nanosheets interlaced with each other that grew vertically on the FTO. After calcination at high temperature, the loosely arranged BiOI nanosheets expose a large specific surface area, which provides enough space for vanadium ions to enter BiOI and convert to $BiVO_4$ [52]. As shown in Figure 1b, after the phase change to $BiVO_4$, the samples transformed from nanosheet morphology to a porous worm-like morphology with a smooth surface that was uniformly dispersed on the FTO glass [53]. The SEM after the doping of Nd is shown in Figure 1c, and it can be clearly observed that the surface of the worm-like structure becomes quite rough and more tightly connected compared with pure $BiVO_4$, and there are many voids between the Nd-$BiVO_4$ nanoparticles, making it easier for the cavities to participate in the water oxidation reaction. It is obvious from Figure 1d and Figure S1 that after hydrothermal treatment, CoOOH is completely wrapped around the Nd-$BiVO_4$ photoanode, and CoOOH-Nd-$BiVO_4$ photoanode is obtained.

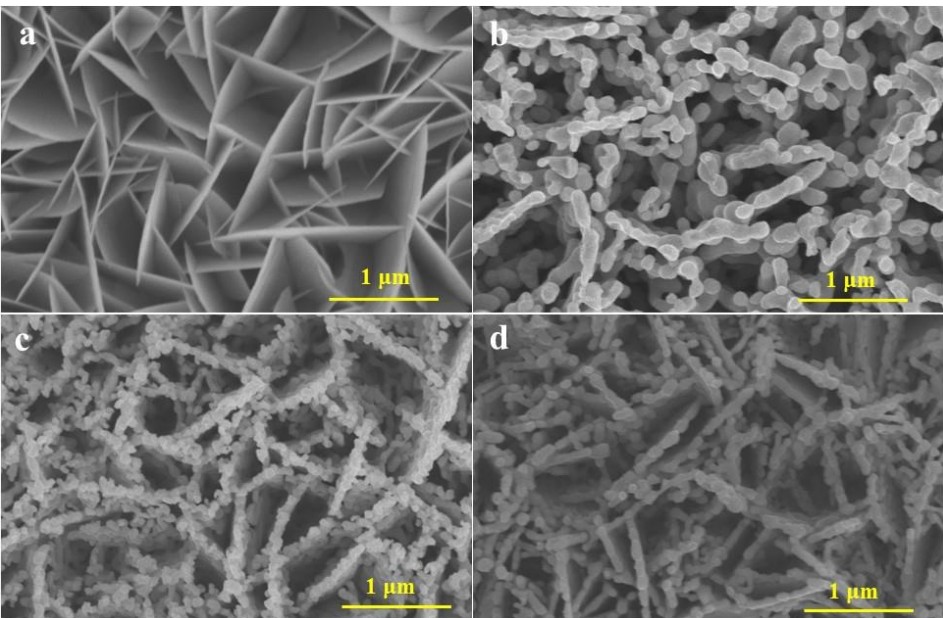

**Figure 1.** SEM of (**a**) BiOI, (**b**) $BiVO_4$, (**c**) Nd-$BiVO_4$, and (**d**) CoOOH-Nd-$BiVO_4$.

The structure and morphology of bare $BiVO_4$ and modified $BiVO_4$ photoanodes were further characterized by transmission electron microscopy (TEM) and high-resolution transmission electron microscopy (HR-TEM). Figure S2 shows the HR-TEM images of $BiVO_4$ and the TEM images of the three photoanodes. The HR-TEM images show that the lattice spacing of $BiVO_4$ is 0.308 nm corresponding to the (1 2 1) crystal plane. Figure 2a,b shows the HR-TEM images of Nd-$BiVO_4$ and CoOOH-Nd-$BiVO_4$, respectively, and it can be clearly seen that doping with rare earth element Nd caused lattice distortion and thus a double lattice appeared [54]. This indicates that Nd doping caused the lattice distortion of $BiVO_4$. After loading CoOOH, as marked by the yellow dashed line, there is a clear interface between Nd-$BiVO_4$ and the cocatalyst, and the average thickness of the amorphous CoOOH layer is measured to be 4 nm, indicating the successful doping of Nd and the successful loading of CoOOH cocatalyst. Figure 2c is the energy dispersive X-ray element

mapping (EDS-mapping) diagram. It can be seen the uniform distribution of elements Bi, V, O, Nd, and Co, indicating that the CoOOH-Nd-BiVO$_4$ composite photoanode was successfully prepared.

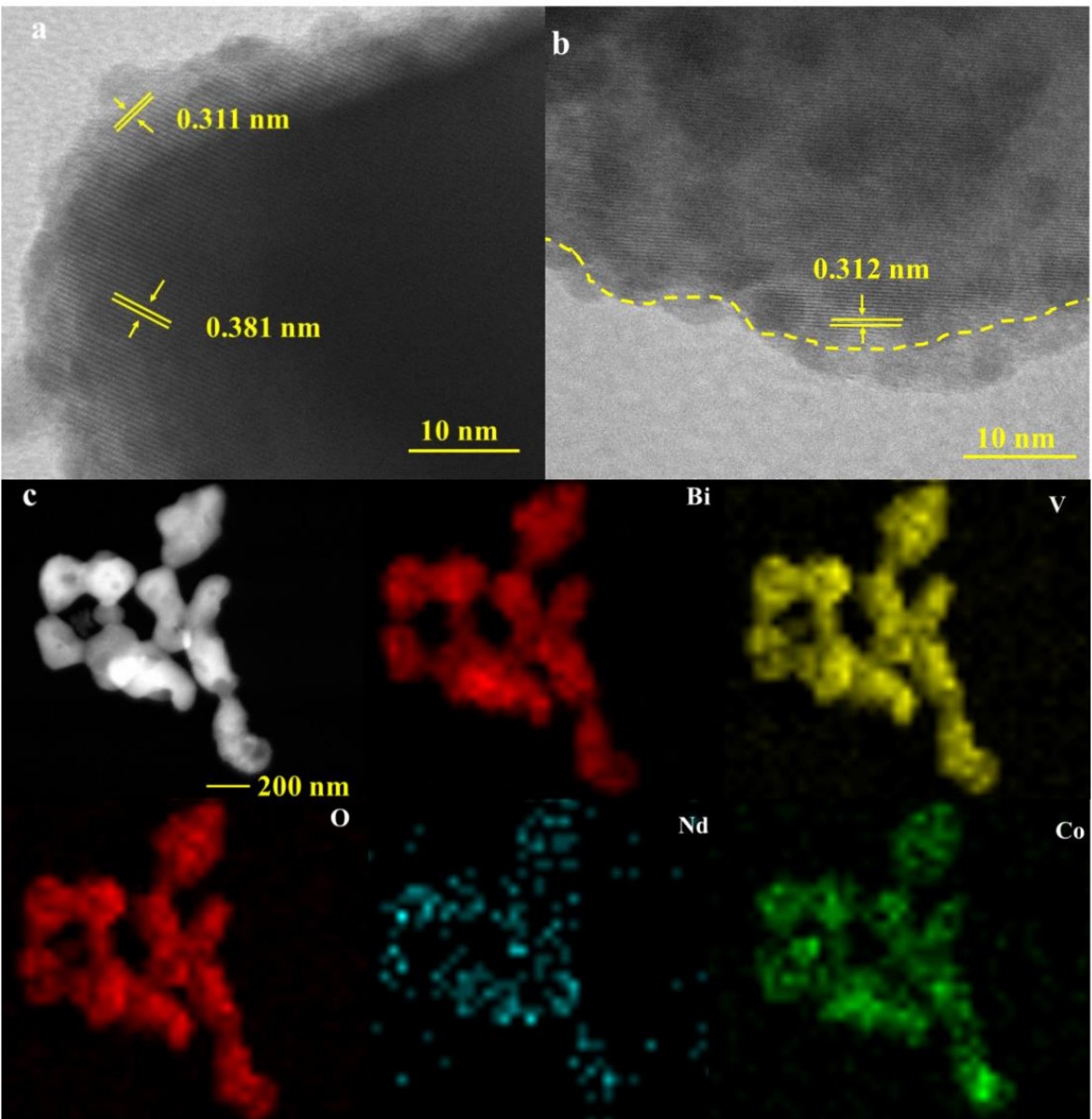

**Figure 2.** (**a**) HR-TEM images of Nd-BiVO$_4$, (**b**) HR-TEM images of CoOOH-Nd-BiVO$_4$, and (**c**) EDS-Mapping images of CoOOH-Nd-BiVO$_4$.

Figure 3 shows the X-ray diffraction (XRD) patterns of bare BiVO$_4$, Nd-BiVO$_4$, and CoOOH-Nd-BiVO$_4$ photoanodes, and it is obvious from the figure that except for the FTO glass diffraction peak (SnO$_2$ PDF 41-1445), the diffraction peaks of bare BiVO$_4$ and doped BiVO$_4$ photoanodes all match the BiVO$_4$ standard card (PDF 14-0688). In other words, Nd doping makes no obvious change on the crystal structure of BiVO$_4$. The CoOOH-Nd-BiVO$_4$ photoanode also shows only the characteristic diffraction peaks of SnO$_2$ and BiVO$_4$, indicating the low CoOOH loading on the surface of the target photoanode as well as the amorphous nature of the loaded CoOOH [55]. This is consistent with the conclusion of HR-TEM.

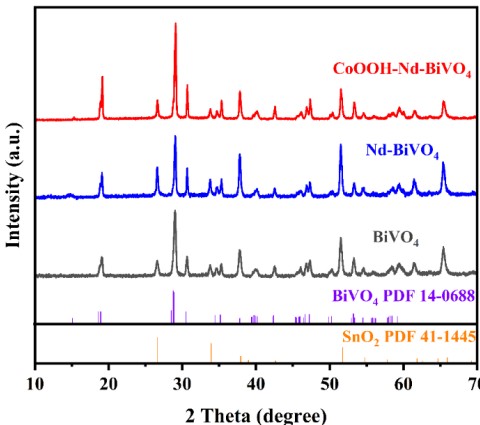

**Figure 3.** XRD patterns of BiVO$_4$, Nd-BiVO$_4$, and CoOOH-Nd-BiVO$_4$.

X-ray photoelectron spectroscopy (XPS) is used to characterize the elemental composition as well as the surface chemical states of photoanode materials, from which the chemical states of Bi, V, O, Nd, and Co can be observed. Figure 4a shows the high-resolution Bi 4f XPS spectrum with peaks at 158.8 eV and 164.1 eV attributed to Bi 4f$_{7/2}$ and Bi 4f$_{5/2}$, respectively [51]. Figure 4b shows the V 2p spectrum with peaks at 516.3 eV and 523.9 eV attributed to V 2p$_{3/2}$ and V 2p$_{1/2}$, respectively [30]. When doped with Nd and loaded with CoOOH, the Bi 4f and V 2p binding energy were both positively shifted. It is possible that Nd replaces V and the loading of CoOOH decreases the electron density, making the binding energy increase. Figure 4c shows the XPS patterns of Nd 3d for doped BiVO$_4$ and target photoanode, indicating that Nd was successfully doped in the BiVO$_4$ photoanode. Figure 4d shows the XPS spectrum of Co 2p, where Co 2p$_{3/2}$ is split into Co$^{3+}$ at 780.7 eV and Co$^{2+}$ at 782.5 eV, accompanied by a satellite peak at 786.2 eV [56]. Figure S3 shows the XPS spectrum of O 1s, where two fitted peaks correspond to lattice oxygen (O$_L$) and hydroxyl oxygen (O$_{OH}$) [57]. For BiVO$_4$ and Nd-BiVO$_4$, the peak at 531.4 eV is ascribed to -OH on account of the deposition of CoOOH.

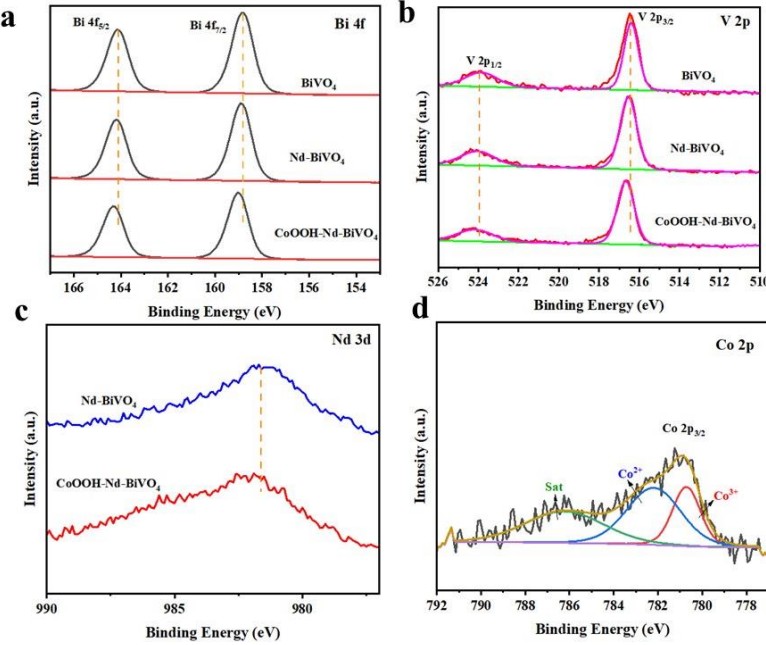

**Figure 4.** XPS patterns of (**a**) Bi 4f, (**b**) V 2p, (**c**) Nd 3d, and (**d**) Co 2p in BiVO$_4$, Nd-BiVO$_4$, and CoOOH-Nd-BiVO$_4$ photoanodes.

To investigate the effects of doped rare earth elements (Nd) and loaded cocatalyst (CoOOH) on the PEC performance of BiVO$_4$ photoanodes, photoelectrochemical tests were performed on all photoanodes. Linear voltammetric scanning (LSV) curves were performed in 0.5 M Na$_2$SO$_4$ solution (pH = 6.4) and under light conditions with a light intensity of 100 mW cm$^{-2}$. As shown in Figure 5a, the photocurrent density of the pure BiVO$_4$ photoanode is 0.9 mA cm$^{-2}$ at 1.23 V vs. RHE. After doping with Nd, the photocurrent density increases to 1.45 mA cm$^{-2}$, because Nd doping may replace the V position, which can cause lattice distortion and the dopant can introduce impurity energy levels in the semiconductor and promote the separation of photogenerated carriers [58]. Meanwhile, the special 4f electron leaping property of Nd can trap electrons and improve the hole separation efficiency [59]. The current density of the CoOOH-Nd-BiVO$_4$ photoanode was significantly enhanced (2.4 mA cm$^{-2}$), which was 2.67 times higher than that of the pure BiVO$_4$ photoanode, and the onset oxygen evolution potential was also significantly negatively shifted by 214 mV. It exhibits that the loading of CoOOH cocatalyst can promote the water oxidation activity of BiVO$_4$ photoanode [55]. Figure 5b shows the LSV curves measured under dark conditions, which can be used to compare the OER activity of the photoanodes: the CoOOH-Nd-BiVO$_4$ photoanodes show the lowest onset potential under dark conditions. Figure 5c shows the LSV curves of all photoanodes under chopped light conditions at 1.23 V vs. RHE, and the values of photocurrent density of all the photoanodes remain consistent with Figure 5a. In addition, the bias photocurrent conversion efficiency (ABPE) of three photoanodes was also calculated to evaluate the photoelectric conversion efficiency of photoanodes (Equation (S2)). As shown in Figure 5d, the maximum ABPE values of bare BiVO$_4$, doped BiVO$_4$, and target photoanodes are 0.07%, 0.11%, and 0.24%, respectively. This suggests that Nd doping and CoOOH modification can greatly increase ABPE.

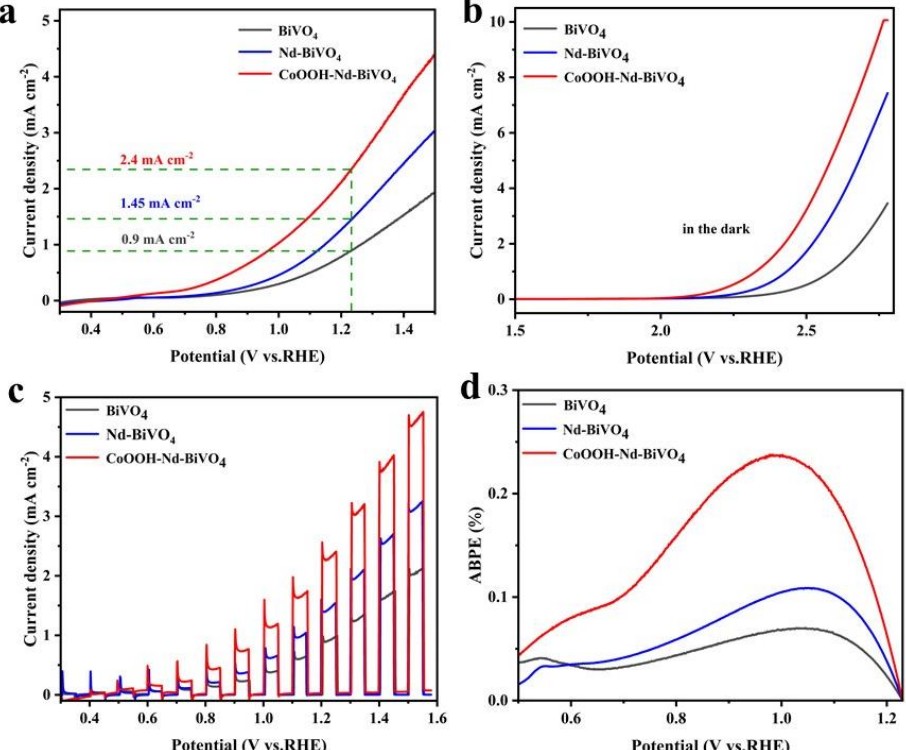

**Figure 5.** (**a**) LSV curves under light, (**b**) LSV curves under dark conditions, (**c**) LSV curves under chopped light conditions, and (**d**) ABPE of BiVO$_4$, Nd−BiVO$_4$, and CoOOH−Nd−BiVO$_4$ photoanodes measured at 1.23 V vs. RHE under AM 1.5 G illumination (100 mW cm$^{-2}$).

In order to further enumerate the working mechanism of photoanode, we carried out the following research. Figure 6a is the Mott–Schottky curve (M–S). By calculating the

slope and intercept of M–S, it is concluded that the flat band potential of the doped and loaded cocatalyst photoanode shifts negatively in turn, and relative to pure $BiVO_4$, the slope is the lowest. This means that the carrier concentration of the composite photoanode $CoOOH$-$Nd$-$BiVO_4$ is greater. The carrier densities of bare $BiVO_4$, doped $BiVO_4$, and the target photoanode are $7.19 \times 10^{20}$, $8.70 \times 10^{20}$, and $1.27 \times 10^{21}$, respectively, derived from Equation (S3). The kinetics of water oxidation of the composite photoanode is investigated using electrochemical impedance spectroscopy (EIS). The EIS data can be fitted by the equivalent circuit diagram in Figure S4. As can be seen from Figure 6b, the radius of $CoOOH$-$Nd$-$BiVO_4$ composite photoanode is the smallest, indicating that the resistance of both composite photoanodes is smaller than that of bare $BiVO_4$. Collectively, the analysis revealed that doping and cocatalyst loading can increase the photocarrier migration of the target photoanode. Figure 6c shows the transient photocurrent curves of the measured photoanodes under alternating light and dark conditions, which can further illustrate the carrier migration behavior of different photoanodes. At the moment of illumination, the photocurrent density decreases significantly because the photogenerated charge inside the photoanode is not exported in time after illumination, which makes the accumulation of electrons and holes be seriously compounded. However, when loaded with $CoOOH$ cocatalyst, this complexation is significantly improved, which is due to the fact that $CoOOH$ can act as a passivation layer and thus inhibit the electron–hole recombination [60]. Analysis of open-circuit photovoltage (OCP) further proves the above conclusion, as shown in Figure 6d, the photovoltage of doped $BiVO_4$ and the target photoanode increases sequentially compared to the unmodified $BiVO_4$.The analysis revealed that the energy band bending at the interface between the photoanode and electrolyte increases, promoting the separation of photogenerated carriers.

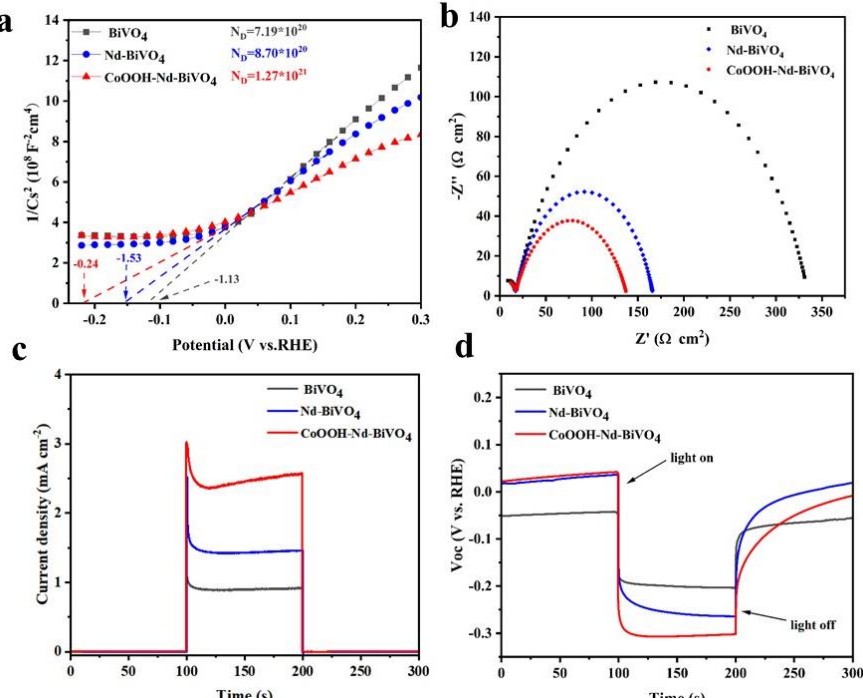

**Figure 6.** (**a**) M−S plots, (**b**) EIS at 1.23 V vs. RHE under light conditions, (**c**) transient photocurrent curves at 1.23 V vs. RHE under light conditions, and (**d**) OCP curves.

The effect of Nd doping and $CoOOH$ cocatalyst on the improved performance of the photoanode was further evaluated by measuring the photocurrent density in the $Na_2SO_4$ electrolyte with $Na_2SO_3$ sacrificial agent. Figure 7a shows the measured LSV curves with $Na_2SO_3$, by which the carrier injection efficiency ($\eta_{surface}$) and carrier separation efficiency ($\eta_{bulk}$) were calculated (Equations (S4)–(S7)). Clearly, Figure 7c shows that the

Nd-BiVO$_4$ (31%) and CoOOH-Nd-BiVO$_4$ (47.5%) composite photoanodes have higher $\eta_{surface}$ at 1.23 V vs. RHE. According to Figure 7d, the $\eta_{bulk}$ of the composite photoanode reaches 78.4% at 1.23 V vs. RHE, which is 13.5% higher relative to the unmodified BiVO4 (65%). This is due to the fact that Nd doping can trap electrons and improve the hole separation efficiency, while the cocatalyst further improves the surface separation efficiency by promoting the hole extraction. It indicates that the doping and the cocatalyst can effectively promote the charge transfer at the interface between the photoanode and the electrolyte, thus accelerating the surface OER kinetics of the water oxidation reaction. The UV-Vis absorption spectra in Figure 7b shows that the absorption band edges of all three photoanodes are around 500 nm. The absorption intensity of the photoanodes is enhanced by the doping of Nd, which is consistent with those reported in the literature [54]. The band gap values of the photoanodes were calculated according to Figure 7b, as shown in Figure S5 (Equation (S8)). The values of the two photoanodes modified with doping and cocatalyst are similar (~2.50 eV), which is consistent with the reported band gap of 2.40–2.50 eV for monoclinic bismuth vanadate [61]. As well, Figure S6 shows that the J$_{abs}$ values of the three photoanodes were calculated according to Equation (S5).

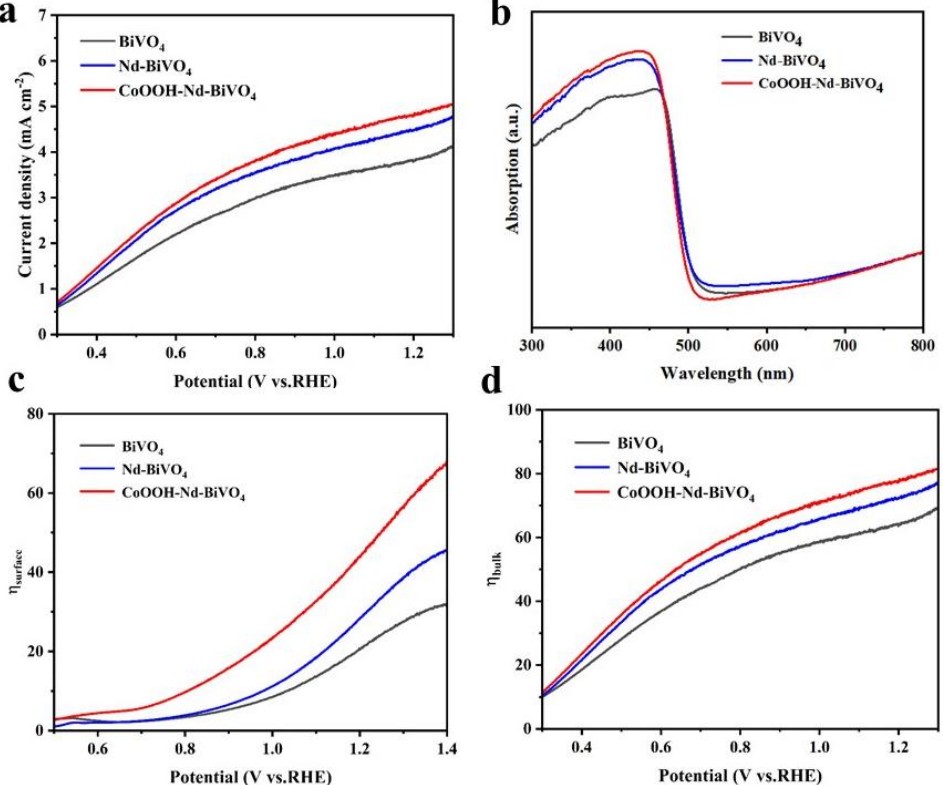

**Figure 7.** (**a**) LSV curves in 0.2 M Na$_2$SO$_3$ sacrificial agent, (**b**) UV−Vis absorption spectra, (**c**) charge injection efficiency, and (**d**) charge separation efficiency for BiVO$_4$, Nd−BiVO$_4$, and CoOOH−Nd−BiVO$_4$ photoanodes.

In addition, the stability of the target photoanode was also tested in 0.5 M Na$_2$SO$_4$ solution. As can be seen in Figure 8, the photocurrent density of the pure BiVO$_4$ photoanode decays by 56% after 7200 s, which is caused by photocorrosion. In contrast, the target photoanode decays only 17% after 7200 s. Its stability is greatly improved because the modification of the CoOOH layer provides a passivation layer to the photoanode material, which makes the photoanode free from photocorrosion [62]. Thus, this modification significantly improves the corrosion resistance of the photoanode and makes it more stable.

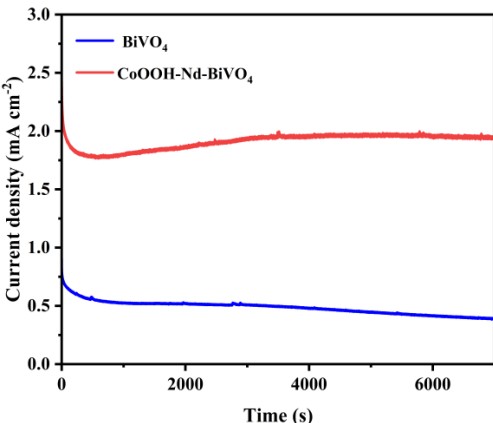

**Figure 8.** Photostability of $BiVO_4$ and $CoOOH-Nd-BiVO_4$ photoanodes.

## 3. Experimental Section

### 3.1. Preparation of $BiVO_4$ and $Nd-BiVO_4$ Photoanodes

The $BiVO_4$ and $Nd-BiVO_4$ photoanodes are prepared by electrochemical deposition [32,63]. The specific steps are described in the supporting information.

### 3.2. Preparation of $CoOOH-Nd-BiVO_4$ Photoanode

The CoOOH layer was modified on the $Nd-BiVO_4$ film by a hydrothermal method [64]. The CoOOH precursor solution containing 5.0 mM $Co(NO_3)_2 \cdot 6H_2O$ (99%), 50.0 mM $NH_4F$ (99.9%), and 50.0 mM urea (99%) was transferred to a 25 mL Teflon lined autoclave, the $Nd-BiVO_4$ photoanode was immersed and tilted to the autoclave wall with the conductive film placed at an angle of 45° with the conductive film facing downward. After hydrothermal treatment at 100 °C for 4 h, it was cooled to room temperature. The composite photoanode was washed with deionized water and ethanol in turn, and then dried with nitrogen gas to obtain $CoOOH-Nd-BiVO_4$ composite photoanode. The process is described in Scheme 1.

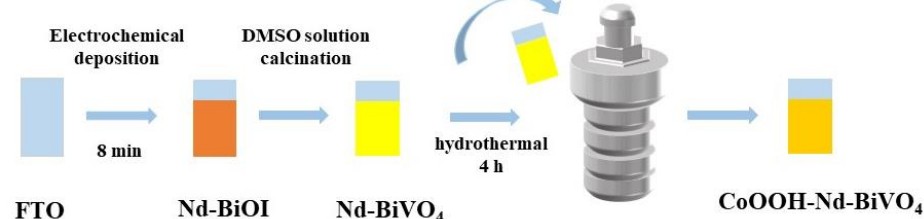

**Scheme 1.** The schematic diagram of the synthesis of $CoOOH-Nd-BiVO_4$ photoanode.

## 4. Conclusions

In summary, a novel photoanode ($CoOOH-Nd-BiVO_4$) material was prepared by rare earth element (Nd) doping and amorphous cocatalyst (CoOOH) modification to improve the water splitting performance of PEC. The Nd doping traps electrons and increase the hole separation efficiency due to lattice distortion, while the modification of amorphous CoOOH cocatalyst accelerates the carrier transfer, inhibits electron–hole recombination, and promotes OER catalytic kinetics. Combining the advantages of Nd doping and CoOOH cocatalyst, the photocurrent density of the $CoOOH-Nd-BiVO_4$ composite photoanode reaches 2.4 mA cm$^{-2}$ at 1.23 V vs. RHE. which is about 2.67 times that of the pure $BiVO_4$ photoanode (0.9 mA cm$^{-2}$). The cocatalyst not only increases the photocurrent density of the material, but also greatly reduces the initial potential of the reaction, which is attributed to the introduction of the cocatalyst to accelerate the oxidation kinetics of the reaction. Therefore, the $CoOOH-Nd-BiVO_4$ composite photoanode has better PEC water oxidation activity, and the doping of rare earth elements and further amorphous cocatalyst loading can provide new ideas for the acceleration of PEC water oxidation reaction.

**Supplementary Materials:** The following supporting information can be downloaded at: https://www.mdpi.com/article/10.3390/inorganics11050203/s1, Figure S1: SEM of CoOOH-Nd-BiVO$_4$; Figure S2: (a) HR-TEM of BiVO$_4$, TEM of (b) BiVO$_4$, (c) Nd-BiVO$_4$, and (d) CoOOH-Nd-BiVO$_4$, respectively; Figure S3: XPS patterns of O 1s in BiVO$_4$, Nd-BiVO$_4$, and CoOOH-Nd-BiVO$_4$ photoanodes; Figure S4: Equivalent electric circuit fitting from EIS curve; Figure S5: Tauc plots of (a) BiVO$_4$, (b) Nd-BiVO$_4$, and (c) CoOOH-Nd-BiVO$_4$ photoanodes; Figure S6: Calculated J$_{abs}$ values of (a) BiVO$_4$, (b) Nd-BiVO$_4$, and (c) CoOOH-Nd-BiVO$_4$ photoanodes. Preparation of BiVO$_4$ and Nd-BiVO$_4$ photoanodes. Materials. Material characterization. Photoelectrochemical test. Calculations (Equations (S1)–(S8)). References [65,66] are cited in Supplementary Materials.

**Author Contributions:** Conceptualization, M.W., L.W. and J.J.; data curation, M.W. and J.G.; formal analysis, M.W., F.Z., L.G. (Lili Gao), L.G. (Lei Geng) and Y.L.; funding acquisition, L.W.; investigation, M.W., L.G. (Lei Geng) and H.N.; methodology, M.W. and K.T.; resources, L.W. and J.J.; supervision, L.W. and J.J.; validation, M.W.; visualization, M.W. and F.Z.; writing—original draft, M.W.; writing—review & editing, M.W., L.W., L.G. (Lili Gao) and H.C. All authors have read and agreed to the published version of the manuscript.

**Funding:** This research was funded by the National Natural Science Foundation of China (No. 31760608), the Key Program of Natural Science Foundation of Gansu Province (No. 22JR5RA178), and the Gansu Key Research and Development Program (No. 22YF7FA172).

**Data Availability Statement:** Data are contained within the article.

**Conflicts of Interest:** The authors declare no conflict of interest.

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
