# Peer review of "Doping with Rare Earth Elements and Loading Cocatalysts to Improve the Solar Water Splitting Performance of BiVO4"

_inorganics, doi:10.3390/inorganics11050203_

Round 1

Reviewer 1 Report

The authors presented a paper on the use of dopants and cocatalysts on BiVO4-based photoanodes for the photoelectrochemical splitting of water. For this purpose, the authors use rare earth elements for doping and Co-based cocatalysts.

The first observation, which arises spontaneously from the materials used is: “Did the authors make a forecast of the costs (per square meter) of possible large-scale use of the device they proposed?”.

The use of bismuth and vanadium in the basic photoanodic semiconductor should also be considered.

In terms of higher photocurrent density does it make sense to use rare earth elements instead of abundant elements and above noncritical raw materials (ref. 16-17)? The literature also reports on MDPI journals’ photocurrent density values higher than those highlighted by the authors and above all at working potentials much lower than 1.23 V.

What are the photoanode active area dimensions prepared and used in the tests? In terms of scalability of the device is a key parameter.

Author Response

Comments and Suggestions for Authors

The authors presented a paper on the use of dopants and cocatalysts on BiVO4-based photoanodes for the photoelectrochemical splitting of water. For this purpose, the authors use rare earth elements for doping and Co-based cocatalysts.

1. The first observation, which arises spontaneously from the materials used is: “Did the authors make a forecast of the costs (per square meter) of possible large-scale use of the device they proposed?”.

Answer: Thanks for your valuable comments on the paper which would help us in depth improve the quality of the paper. Since the stability of BiVO4 is one of the issues for large-scale use, it is still in a research stage. Considering the production cost, with the continuous development of science and technology, the use of earth-rich, cheap and effective rare earth elements has been continuously developed, as mentioned in the literature (Accounts of Chemical Research, 2018, 51(11), 2926-2936, https://doi.org/10.1021/acs.accounts.8b00336). The same is true of the rare earth elements (Nd) used.

2. The use of bismuth and vanadium in the basic photoanodic semiconductor should also be considered.

Answer: Thanks for your valuable comments on the paper which would help us in depth improve the quality of the paper. As stated in the literature, in the process of preparing bismuth vanadate photoanode, electrochemical deposition method is used (Science, 2014, 343(6174): 990-994, https://doi.org/10.1126/science.1246913). The BiVO4 prepared by this method has good photoelectric effect and has been widely used. The reagent used is VO(acac)2 as a common organic metal compound is widely used as oxidation catalyst, catalyst precursor, medicine, coating desiccant, pigment and so on. Bismuth nitrate is an important bismuth (â…¢) salt and is widely used in practical work.

3. In terms of higher photocurrent density does it make sense to use rare earth elements instead of abundant elements and above noncritical raw materials (ref. 16-17)? The literature also reports on MDPI journals’ photocurrent density values higher than those highlighted by the authors and above all at working potentials much lower than 1.23 V.

Answer: Thanks for your valuable comments on the paper which would help us in depth improve the quality of the paper. Table R1 compares CoOOH-Nd-BiVO4 photoanodes with other BiVO4 composite photoanodes in terms of current density, synthesis methods and stability. The results show that the CoOOH-Nd-BiVO4 photoanode synthesized by this method has good current density and excellent stability.

Table R1 Current density of different composite photoanodes

Catalyst

Dopant

Current density

at 1.23

VRHE (mA cm-2)

Stability

Preparation method

Ref.

CoOOH

Nd

2.4

About 17% loss after 7200 seconds of continuous operation

Electroplating method + Hydrothermal synthesis method

This work

FeOOH

W

2.2

About 20% loss after 3000 seconds of continuous operation

photoelectrochemical technique + photoelectrochemical technique

International Journal of Hydrogen Energy, 2022, 47, 27012-27022, https://doi.org/10.1016/j.ijhydene.2022.06.048

NiFe2O4

Nd

1.93

About 30% loss after 8 hours of continuous operation

Electroplating method + Spin coating calcination method

Journal of Alloys and Compounds, 2022, 923, 166352, https://doi.org/10.1016/j.jallcom.2022.166352

NH2-MIL-88B(Fe)

Mo

1.46

About 6% loss after 1000 seconds of continuous operation

Electroplating method + Hydrothermal synthesis method

Journal of Photochemistry and Photobiology A: Chemistry, 2022, 431, 114049, https://doi.org/10.1016/j.jphotochem.2022.114049

MIL-53 (Fe)

Fe

2.52

About 15% loss after 600 seconds of continuous operation

Drip calcination method + Spin coating calcination method

Catalysis Letters, 2019, 149, 870-875, https://doi.org/10.1007/s10562-018-2629-4

NiFe-LDH

Mo

1.58

About 42% loss after 8000 seconds of continuous operation

 Metal organic decomposition technique + Electroplating method

Journal of Colloid and Interface Science, 2019, 540, 9-19, https://doi.org/ 10.1016/j.jcis.2018.12.069

4. What are the photoanode active area dimensions prepared and used in the tests? In terms of scalability of the device is a key parameter.

Answer: Thanks for your valuable comments on the paper which would help us in depth improve the quality of the paper. We have carefully revised the manuscript according to your advice. The photoanode activity area used for testing was 1x1 cm2, details are added to the photoelectrochemical test testing section of the Supporting Information.

Reviewer 2 Report

Minor Revision

(inorganics-2343450)

The submitted manuscript (inorganics-2343450) “Doping with rare earth elements and loading cocatalysts to improve the solar water splitting performance of BiVO4” This article focused on BiVO4 which is a highly promising material for PEC water splitting photoanodes due to its narrow band gap value (~2.4 eV) and its ability to efficiently absorb visible light. The author claimed that the prepared target photoanode (CoOOH-Nd-BiVO4) exhibits an excellent photocurrent density (2.4 mAcm-2) at 1.23 V versus reversible hydrogen electrode potential (vs. RHE), which is 2.67 times higher than that of pure BiVO4 (0.9 mA cm-2), and the onset potential is negatively shifted by 214 mV.

However, the results and explanation are reasonable but still needs to be revised with minor errors before accepted for publication.  

Reviewer comment;

1. In abstract: At the end, the author needs to mention the novel statement of this work.

2. Materials and Methods section is missing; the author includes this section with the purity % of used precursors.

3. The detailed of instrumentation analysis also missing, needs to be included with proper explanation and condition of instruments used for analysis.

4. Regarding originality and scientific progress of this work, in the introduction section, author needs to include more information and mention the advantages of other metal oxides over BiVO4.

5. In introduction section, author did not mention information regarding properties of rare earth materials. It is recommended to read the given article, add information and cite the given articles.

6. The authors need to include the advantage and cost-effectiveness of BiVO4 decorated with rare earth elements utilizes in water splitting performance.

7. Figure 1, could you use the real scale for figures (SEM a-d).

8. The lattice fringes is not clear where author highlighted with yellow marker, use the clear image or remove the highlights.

9. In XPS analysis, please provide the survey scan spectra for rare earth metal doping confirmation.

10. The author should make the necessary modifications, addition, citation and responses to the queries raised above before acceptance.

Author Response

Comments and Suggestions for Authors

Minor Revision

(inorganics-2343450)

The submitted manuscript (inorganics-2343450) “Doping with rare earth elements and loading cocatalysts to improve the solar water splitting performance of BiVO4” This article focused on BiVO4 which is a highly promising material for PEC water splitting photoanodes due to its narrow band gap value (~2.4 eV) and its ability to efficiently absorb visible light. The author claimed that the prepared target photoanode (CoOOH-Nd-BiVO4) exhibits an excellent photocurrent density (2.4 mAcm-2) at 1.23 V versus reversible hydrogen electrode potential (vs. RHE), which is 2.67 times higher than that of pure BiVO4 (0.9 mA cm-2), and the onset potential is negatively shifted by 214 mV.

However, the results and explanation are reasonable but still needs to be revised with minor errors before accepted for publication.  

Reviewer comment;

1. In abstract: At the end, the author needs to mention the novel statement of this work.

Answer: Thanks for your valuable comments on the paper which would help us in depth improve the quality of the paper. We have carefully revised the manuscript according to your advice. The details are as follows: The formation of the internal energy states of rare earth metal elements can reduce the photoexcited electron-hole pair recombination, so as to achieve efficient photochemical water decomposition ability. CoOOH is an efficient and suitable oxygen evolution cocatalyst (OEC), and OEC decoration of BiVO4 surface is of great significance for inhibiting surface charge recombination.

2. Materials and Methods section is missing; the author includes this section with the purity % of used precursors.

Answer: Thanks for your valuable comments on the paper which would help us in depth improve the quality of the paper. We have carefully revised the manuscript according to your advice. The details are as follows: The CoOOH precursor solution containing 5.0 mM Co(NO3)2⋅6H2O (99%), 50.0 mM NH4F (99.9%) and 50.0 mM urea (99%) was transferred to a 25 mL Teflon lined autoclave.

3. The detailed of instrumentation analysis also missing, needs to be included with proper explanation and condition of instruments used for analysis.

Answer: Thanks for your valuable comments on the paper which would help us in depth improve the quality of the paper. We have carefully revised the manuscript according to your advice. The detailed of instrumentation analysis is mentioned in the Supporting Information.

4. Regarding originality and scientific progress of this work, in the introduction section, author needs to include more information and mention the advantages of other metal oxides over BiVO4.

Answer: Thanks for your valuable comments on the paper which would help us in depth improve the quality of the paper. We have carefully revised the manuscript according to your advice. The details are as follows: Both TiO2 and ZnO have wide energy band gaps (>3.0 eV), which makes them only absorb ultraviolet light, resulting in extremely low solar energy utilization and hydrogen production efficiency. Fe2O3 has poor conductivity, low light absorption coefficient, short hole diffusion length (2-4 nm), and poor surface oxygen evolution kinetics. WO3 has poor light absorption ability and is thermodynamically unstable in the electrolyte, which is susceptible to photocorrosion caused by peroxide species generated during water oxidation.

5. In introduction section, author did not mention information regarding properties of rare earth materials. It is recommended to read the given article, add information and cite the given articles.

Answer: Thanks for your valuable comments on the paper which would help us in depth improve the quality of the paper. We have carefully revised the manuscript according to your advice. The details are as follows: The rare earth element has a special 4f orbital, and its ion configuration is 4fn5s25p6. We also cite the relevant literature (Trends in Chemistry 2019, 1, 751-762, https://doi.org/10.1016/j.trechm.2019.05.012). Therefore, this makes them have unique chemical properties and a wider space for use.

6. The authors need to include the advantage and cost-effectiveness of BiVO4decorated with rare earth elements utilizes in water splitting performance.

Answer: Thanks for your valuable comments on the paper which would help us in depth improve the quality of the paper. We have carefully revised the manuscript according to your advice. The details are as follows: with the continuous development of science and technology, the use of earth-rich, cheap and effective rare earth element photocatalysts has been continuously developed. This will have the opportunity to replace rare and precious metals. Umesh Prasad et al.reported a BiVO4 doped with Er, W, and constructed a heterojunction photoanode with WO3, namely WO3/(Er, W): BiVO4 photoanode, which enhanced PEC performance. The rare earth element Er-doped BiVO4 can improve the bulk charge separation efficiency, thereby improving the conductivity of the charge carriers.

7.  Figure 1, could you use the real scale for figures (SEM a-d).

Answer: Thanks for your valuable comments on the paper which would help us in depth improve the quality of the paper. The real scale of SEM (a-d) is shown in Figure R1, which is consistent with the scale of Figure 1 in the manuscript. For the sake of aesthetics and better presentation of SEM size, we simulate the real SEM scale.

Fig. R1 SEM of (a) BiOI, (b) BiVO4, (c) Nd-BiVO4, (d) CoOOH-Nd-BiVO4.

8. The lattice fringes is not clear where author highlighted with yellow marker, use the clear image or remove the highlights.

Answer: Thanks for your valuable comments on the paper which would help us in depth improve the quality of the paper. We have carefully revised the manuscript according to your advice. See the following chart for details.

Fig. R2 (a) HR-TEM images of Nd-BiVO4 and (b) HR-TEM images of CoOOH-Nd-BiVO4, (c) EDS-Mapping images of CoOOH-Nd-BiVO4.

9. In XPS analysis, please provide the survey scan spectra for rare earth metal doping confirmation.

Answer: Thanks for your valuable comments on the paper which would help us in depth improve the quality of the paper. We have carefully revised the manuscript according to your advice. We measured the XPS spectrum of CoOOH-Nd-BiVO4 photoanode, as shown in Fig. R3, and the characteristic peaks of Bi, V, O, Nd and Co elements can be seen from the XPS full spectrum. And the atomic ratios of O, Bi, V, Nd and Co elements were derived from the XPS full spectrum.

Fig. R3 XPS full spectrum of CoOOH-Nd-BiVO4 photoanode

10. The author should make the necessary modifications, addition, citation and responses to the queries raised above before acceptance.

Answer: Thanks for your valuable comments on the paper which would help us in depth improve the quality of the paper. We have carefully revised the manuscript according to your advice.

Reviewer 3 Report

The manuscript of Wang et al. attempts to elucidate the influence of Nd doping and CoOOH cocatalyst on the photoelectrochemical H2O splitting activity of BiVO4. Even though the idea is not so novel, the authors comprehensively studied and analyzed the results. Moreover, a lot of improvements are needed before it can be recommended for publication.

1. Please provide proof that the existing Co species is CoOOH and not in other oxide form. It would be better if post-OER characterizations are also done to confirm if the cocatalyst did not change its structure after catalysis. This is a crucial part for reproducibility issues, as critically discussed in this reference: Chem. Sci., 2022,13, 2824-2840. If not possible, please add a discussion on the importance of structural and chemical transformation of cocatalysts during photo(electro)catalysis.

2. How much Nd is doped onto BiVO4 and what is the loading amount of CoOOH with respect to BiVO4?

3. Could you provide more than 1 cycle for transient photocurrent response in order to show the recyclability of the photoanodes?

4. The presence of lattice distortion and double lattice is not obvious from the TEM images. Could you provide higher resolution TEM and more lattice spacing measurements for accuracy? Based on XRD, there is no detected peak shift. 

5. It seems that the CoOOH cocatalyst is covering the BiVO4 photoanode, did you try to optimize the loading of CoOOH to get a more desirable distribution of the cocatalyst?

Author Response

Comments and Suggestions for Authors

The manuscript of Wang et al. attempts to elucidate the influence of Nd doping and CoOOH cocatalyst on the photoelectrochemical H2O splitting activity of BiVO4. Even though the idea is not so novel, the authors comprehensively studied and analyzed the results. Moreover, a lot of improvements are needed before it can be recommended for publication.

1. Please provide proof that the existing Co species is CoOOH and not in other oxide form. It would be better if post-OER characterizations are also done to confirm if the cocatalyst did not change its structure after catalysis. This is a crucial part for reproducibility issues, as critically discussed in this reference: Chem. Sci., 2022,13, 2824-2840. If not possible, please add a discussion on the importance of structural and chemical transformation of cocatalysts during photo(electro)catalysis.

Answer: Thanks for your valuable comments on the paper which would help us in depth improve the quality of the paper. Because CoOOH is amorphous, the characteristic peak of CoOOH is not detected in XRD. In TEM images, there is a clear interface between Nd-BiVO4 and the catalyst, and the amorphous CoOOH layer is observed. Secondly, Figure S3 shows the XPS spectrum of O 1s, where two fitted peaks appear in the image corresponding to lattice oxygen (OL) and hydroxyl oxygen (OOH). The successful loading of CoOOH was proved, and the area of OOH in CoOOH-Nd-BiVO4 was significantly increased. It indicates that the existing Co species is CoOOH, not other oxide forms. Moreover, the preparation method of CoOOH is referred to the literature and relevant citations are made (ACS Applied Energy Materials 2022, 5, 11271-11282, https://doi.org/10.1021/acsaem.2c01827). Co3+ is a hole medium for water oxidation and plays a role in transferring holes. According to the Fig. R1, we analyzed the content of trivalent Co cations after PEC test, and the results showed that the concentration ratio of Co3+ decreased from 25.24% to 20.99%. As mentioned in the literature (Nanoscale, 2022, 14(45): 17044-17052, https://doi.org/10.1039/d2nr04445e), it was confirmed that a part of Co3+ was reduced to Co2+ during water oxidation. The proportion of OOH and OL did not change. Therefore, it is further explained that after the PEC test, the recombined oxide exists in the form of hydroxyl oxide.

Fig. R1 XPS patterns of (a, b) O 1s of CoOOH-Nd-BiVO4 photoanode before and after test, (c, d) Co2p of CoOOH-Nd-BiVO4 photoanode before and after test.

2. How much Nd is doped onto BiVO4 and what is the loading amount of CoOOH with respect to BiVO4?

Answer: Thanks for your valuable comments on the paper which would help us in depth improve the quality of the paper. We have carefully revised the manuscript according to your advice. We measured the XPS spectrum of CoOOH-Nd-BiVO4 photoanode, as shown in Fig. R2, and the characteristic peaks of Bi, V, O, Nd and Co elements can be seen from the XPS full spectrum. And the atomic ratios of O, Bi, V, Nd and Co elements were derived from the XPS full spectrum.

Fig. R2 XPS full spectrum of CoOOH-Nd-BiVO4 photoanode

3. Could you provide more than 1 cycle for transient photocurrent response in order to show the recyclability of the photoanodes?

Answer: Thanks for your valuable comments on the paper which would help us in depth improve the quality of the paper. As shown in Fig. R3, We tested the transient photocurrent of the photoanode for two cycles, the photocurrent density of the target photoanode does not decrease after two cycles, indicating that the target photoanode has good repeatability.

Fig. R3 Transient photocurrent of two cycles of CoOOH-Nd-BiVO4 photoanode

4. The presence of lattice distortion and double lattice is not obvious from the TEM images. Could you provide higher resolution TEM and more lattice spacing measurements for accuracy? Based on XRD, there is no detected peak shift. 

Answer: Thanks for your valuable comments on the paper which would help us in depth improve the quality of the paper. We have carefully revised the manuscript according to your advice. See the following chart for details.

Fig. R4. (a) HR-TEM images of Nd-BiVO4 and (b) HR-TEM images of CoOOH-Nd-BiVO4, (c) EDS-Mapping images of CoOOH-Nd-BiVO.

5. It seems that the CoOOH cocatalyst is covering the BiVO4 photoanode, did you try to optimize the loading of CoOOH to get a more desirable distribution of the cocatalyst?

Answer: Thanks for your valuable comments on the paper which would help us in depth improve the quality of the paper. In this work, the target photoanode was optimized by controlling the concentration of Co(NO3)2⋅6H2O. As shown in Fig. R5, the linear scanning voltammetry (LSV) curve was used to evaluate the photoelectrochemical performance.

Fig. R5 LSV curves of composite photoanodes with different Co(NO3)2⋅6H2O concentrations

Round 2

Reviewer 1 Report

The authors presented a paper on the use of dopants and cocatalysts on BiVO4-based photoanodes for the photoelectrochemical splitting of water. For this purpose, the authors use rare earth elements for doping and Co-based cocatalysts.

The authors promptly responded to the reviewer's requests, supporting their answers with appropriate references.